# Efficient Semi-Supervised Multi-Organ Segmentation Using Uncertainty Rectified Pyramid Consistency

Meng Han[1], Yijie Qu[1], and Xiangde Luo[1]

School of Mechanical and Electrical Engineering, University of Electronic Science and Technology of China, Chengdu, China

**Abstract.** To meet the problems that great dependence on fully annotated data and spatio-temporal inefficiency of remaining automatic multi-organ segmentation methods, an efficient semi-supervised framework with uncertainty rectified pyramid consistency regularization is introduced. Specifically, inspired by the fact that the predictions of the same input should be similar under different disturbance, we extend a backbone to produce predictions at different scale for unlabeled images and encourage them to be consistent. Since the multi-scale predictions have different resolution, directly encouraging them to be consistent may bring problems including lost of fine detail or model collapse. So a rectified scale-level uncertainty-aware module is introduced to enable the framework to gradually learn from reliable prediction regions. To deal with the domain gaps among multi-center datasets, a number of prepocessing methods are utilized, such as resampling the multi-center CT volumes to the same spacing and adjusting the window level and width. Quantitative evaluation on the FLARE2022 20 validation cases, this method achieves the average dice similarity coefficient (DSC) of 0.793 and average normalized surface distance (NSD) of 0.852.

**Keywords:** Semi-supervised learning · Multi-organ segmentation · Uncertainty rectifying · Pyramid consistency.

## 1 Introduction

Whole abdominal organ segmentation plays an important role in the diagnosis of abdominal lesions, radiotherapy and follow-up. Manual organ delineation is time-consuming and error-prone[7]. Although many automatic segmentation methods based on deep learning have achieved good performances in abdominal organ segmentation, most of them rely heavily on large-scale fully annotated data, which is often difficult to obtain due to cost and privacy issues. Additionally, these methods are often difficult to be implemented into clinical practice due to the large model size and the extensive computational resources.

To meet the needs of fast inference and low computational cost while only use a small number of labeled cases, we develop an efficient semi-supervised framework with uncertainty rectified pyramid consistency to fully make use of the

unlabeled data. Concretely, we extend a U-Net[9] backbone to produce pyramid predictions at different scales and encourages these multi-scale predictions to be consistent for a given input. A standard supervised loss is used for learning from labeled data. For unlabeled cases, we encourages the pyramid predictions to be consistent, which served as a regularization. Since the predictions of the unlabeled cases are not necessarily reliable because of its insufficient supervision information, which may cause the model to collapse and lose details[6], we propose to estimate the uncertainty via the prediction discrepancy among multi-scale predictions. Different from those that estimate the uncertainty of each target prediction with Monte Carlo sampling[5], which needs massive computational costs as it requires multiple forward passes to obtain the uncertainty in each iteration, our proposal just needs a single forward pass. Under the guidance of the estimated uncertainty, the model strengthens the consistent learning of reliable regions and weakens that of unreliable ones. Meanwhile, to further improve the utilization of unlabeled data information and increase the segmentation efficiency, we introduce the uncertainty minimization[11] to reduce the prediction variance during training.

To address the domain gaps among FLARE2022 training, validation, and test sets, we employ a series of data preprocessing methods. For instance, resampling the images' spacing to $2 \times 2 \times 2.5$mm, adjusting the window level and window width of the CT images, data augmentation with CLAHE, gamma correction, random noise, random rotate and random flip.

## 2   Method

The overview of the proposed semi-supervised segmentation network is illustrated in Fig.1, which consists of a 3D U-Net[9] backbone with a pyramid prediction structure at the decoder and an uncertainty rectifying module.

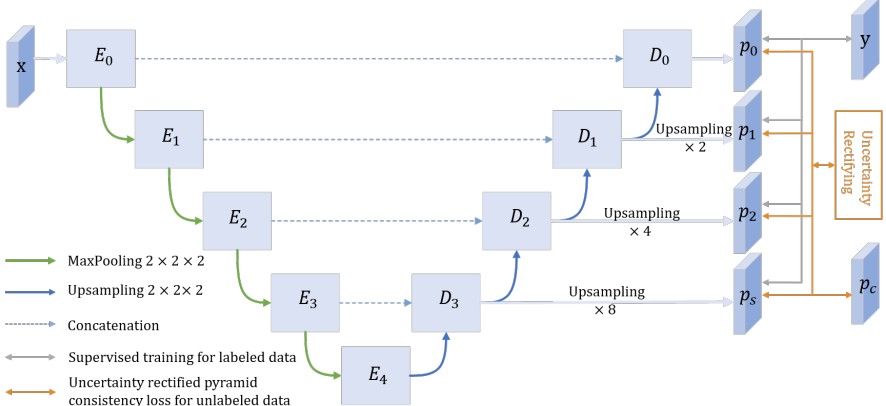

**Fig. 1.** Network architecture.

### 2.1   Multi-Scale Prediction Network with Pyramid Consistency

Firstly, we introduce pyramid prediction network (PPNet) for the multi-organ segmentation task, which can produce prediction at different scales. In this work, a 3D U-Net is employed as backbone and is modified to produce pyramid predictions by adding a prediction layer after each upsampling block in the decoder, where the prediction layer is implemented by $1 \times 1 \times 1$ convolution followed by a softmax layer. A dropout layer and a feature-level noise addition layer are inserted before the prediction layer to introduce more perturbations to the network.

For an input image $x_i$, the PPNet produces a set of multi-scale predictions $[p'_0, p'_1, \cdots, p'_s, \cdots, p'_{S-1}]$, where $S$ means the number of the scales, $p'_s$ means the prediction at scale $s$. A smaller $s$ presents a higher resolution in the decoder. In our work, we set $S = 4$ for the multi-organ segmentation. Then, the multi-scale predictions are rescaled to the input size, and the the corresponding results are denoted as $[p'_0, p_1, \cdots, p_s, \cdots, p_{S-1}]$. For the labeled data, we use a combination of Dice and cross-entropy loss at multiple scales as the supervised loss:

$$\mathcal{L}_{sup} = \frac{1}{S} \sum_{s=0}^{S-1} \frac{\mathcal{L}_{dice}(p_s, y_i) + \mathcal{L}_{ce}(p_s, y_i)}{2} \tag{1}$$

where $y_i, \mathcal{L}_{dice}, \mathcal{L}_{ce}$ denote the ground truth of input $x_i$, the Dice loss and the cross entropy loss, respectively.

To efficiently leverage unlabeled data, we design a pyramid consistency loss to encouraging the multi-scale predictions to be consistent by minimizing their discrepancy(i.e., variance) with $L_2$ distance:

$$\mathcal{L}_{pyc} = \frac{1}{S} \sum_{s=0}^{S-1} \|p_s - p_c\|_2 \tag{2}$$

where $p_c$ is the average prediction across scales, which is denoted as:

$$p_c = \frac{1}{S} \sum_{s=0}^{S-1} p_s \tag{3}$$

### 2.2   Uncertainty Rectified Pyramid Consistency Loss

As the pyramid predictions have different spatial resolutions, imposing voxel-level consistency directly between these predictions may lead to problems due the different spatial frequencies, such as lost of fine detail or model collapse. Inspired by existing works [1,11], a scale-level uncertainty-aware method is introduced to address these problems, which only requires a single forward pass and thus need less computational cost and running time than exiting methods[5].

We use the KL-divergence between the prediction at scale $s$ and the mean prediction as the uncertainty measurement:

$$\mathcal{D}_s \approx \sum_{j=0}^{C} p_s^j \cdot \log \frac{p_s^j}{p_c^j} \tag{4}$$

where $C = 14$ is the class(i.e., channel) number in this work, and $p_s^j$ is the $j$th channel of $p_s$. We can get a set of uncertainty maps $\mathcal{D}_0, \mathcal{D}_1, \cdots, \mathcal{D}_s, \cdots, \mathcal{D}_{S-1}$, where $\mathcal{D}_s$ represents the uncertainty of $p_s$. A larger value of $\mathcal{D}_s$ indicates the prediction for that pixel at scale $s$ has high uncertainty, which means the prediction is unreliable and may be ignored for stable unsupervised learning.

Based on the uncertainty maps, we automatically select reliable voxels(with low uncertainty) to rectify the pyramid consistency loss for better using the information of unlabeled data:

$$\mathcal{L}_{unsup} = \frac{1}{S} \underbrace{\frac{\sum_{s=0}^{S-1} \sum_v (p_s^v - p_c^v)^2 \cdot w_s^v}{\sum_{s=0}^{S-1} \sum_v w_s^v}}_{uncertainty rectification} + \underbrace{\frac{1}{S} \sum_{s=0}^{S-1} \|\mathcal{D}_s\|_2}_{uncertainty minimization} \tag{5}$$

where $p_s^v$ and $\mathcal{D}_s^v$ are the corresponding prediction and uncertainty values for voxel v. Rather than use the threshold-based cut off approaches which is hard to determine[1], we follow the policy in [11] to use a voxel- and scale-wise weight $w_s^v$ to automatically rectify the MSE loss, which is defined as: $w_s^v = e^{-\mathcal{D}_s^v}$, where a higher uncertainty leads to lower weight. Meanwhile, in order to encourage the PPNet to produce more consistent predictions at different scales, we use the uncertainty minimization term as a constraint.

### 2.3    The Overall Loss Function

The proposed semi-supervised segmentation network learns form both labeled data and unlabeled data by minimizing the following combined objective function:

$$\mathcal{L}_{total} = \mathcal{L}_{sup} + \lambda \cdot \mathcal{L}_{unsup} \tag{6}$$

where $\lambda$ is a widely-used time-dependent Gaussian warming up function [5] to control the balance between the supervised loss and the unsupervised loss. The formula of $\lambda(t)$ is: $\lambda(t) = w_{max} \cdot e^{(-5(1 - \frac{1}{t_{max}})^2}$, where $w_{max}$ means the final regularization weight, $t$ denotes the current training step and $t_{max}$ is the maximal training step.

### 2.4    Preprocessing

To reduce the domain gaps between multi-center data, the following preprocessing techniques are employed:

– Resampling the anisotropic data:

   Due to differences in scanners or acquisition protocols, data from different centers usually have different spacing. The convolution operation of CNN requires the image to be isotropic for better feature extraction. So all data, including training, validation and test data, need to be resampled to keep the image isotropic. Observing the resolution information of the training data, combined with the consideration of the trade-off between the amount of

contextual information in the networks patch size and the details retained in the image data, the sampling layer spacing is set to 2.5mm. To save memory overhead and speed up training, the intra-layer resolution is resampled to 2mm × 2mm.

– Adjusting window level and window width:
  The method of adjusting the window width and window level was leveraged to improve the contrast of abdominal CT images. And the window level and width were adjusted 50 and 400, respectively.
– Intensity normalization method:
  The data were normalized with z-score normalization based on the mean and standard deviation of the intensity to avoid the problem of data being compressed after normalization.
– Data augmentation method:
  Due to the lack of label data and to avoid the problem of overfitting, channel-wise gamma correction, random noise, random flip, random rotate were used to augment the training data.

### 2.5 Proposed Method

– Network architecture details: The network architecture is shown in Fig.1.
– Loss function: the loss function is shown as Formula 6
– Number of the parameters of PPNet: 1330104.

### 2.6 Post-processing

A connected component analysis of segmentation mask is applied on the outputs to remove small connected areas. And then the results are resampled back to original spacing for the convenience of the following evaluation.

## 3 Experiments

### 3.1 Dataset and evaluation measures

The FLARE2022 dataset is curated from more than 20 medical groups under the license permission, including MSD [10], KiTS [3,4], AbdomenCT-1K [8], and TCIA [2]. The training set includes 50 labelled CT scans with pancreas disease and 2000 unlabelled CT scans with liver, kidney, spleen, or pancreas diseases. The validation set includes 50 CT scans with liver, kidney, spleen, or pancreas diseases. The testing set includes 200 CT scans where 100 cases has liver, kidney, spleen, or pancreas diseases and the other 100 cases has uterine corpus endometrial, urothelial bladder, stomach, sarcomas, or ovarian diseases. All the CT scans only have image information and the center information is not available.

The evaluation measures consist of two accuracy measures: Dice Similarity Coefficient (DSC) and Normalized Surface Dice (NSD), and three running efficiency measures: running time, area under GPU memory-time curve(AUC GPU), and area under CPU utilization-time curve(AUC CPU). Moreover, the GPU memory consumption has a 2 GB tolerance.

## 3.2   Implementation details

**Environment settings** The development environments and requirements are presented in Table 1.

**Table 1.** Development environments and requirements.

| | |
|---|---|
| Windows/Ubuntu version | Ubuntu 20.04.4 LTS |
| CPU | Intel(R) Xeon(R) CPU E5-2678 v3 @ 2.50GHz |
| RAM | 16×4GB; 2.67MT/s |
| GPU (number and type) | Four NVIDIA Corporation TU102(2080Ti) 10G |
| CUDA version | 11.4.48 |
| Programming language | Python 3.7.13 |
| Deep learning framework | Pytorch (Torch 1.11.0, torchvision 0.12.0) |
| Specific dependencies | None |

**Training protocols** The training protocols of the baseline method is shown in Table 2.

**Table 2.** Training protocols.

| | |
|---|---|
| Network initialization | "he" normal initialization |
| Batch size | 4 |
| Patch size | 80×96×96 |
| Total epochs | 100 |
| Optimizer | Adam with momentum ($\mu = 0.9$) |
| Initial learning rate (lr) | 0.001 |
| Lr decay schedule | halved by 20 epochs |
| Training time | 24 hours |
| Number of model parameters | 15.3M |
| Number of flops | 59.32G |

## 4   Results and discussion

### 4.1   Quantitative results on validation set

Table 3 illustrates the results of this work on the 20 validation cases whose ground truth are publicly provided by FLARE2022. Large-volume organs like liver, Aorta, spleen and kidney have relatively good performance. However, the

poor performance on small organs and diseased organs reduces the overall average performance. Indeed, the segmentation results have the problem of organ disappearance, that is, the segmentation results of some organs, especially small organs, are not predicted. Another problem is that due to the deformation of some diseased organs as well as the lack training of model on such data, the segmentation prediction of diseased organs is completely wrong. The above two situations are the main reasons that lead to the corresponding DSC and NSD of some cases' organs segmentation valuing 0, which is also a part to be further explored and sovled. It is noted that the values of NSD are generally better than that of DSC, indicating that the method proposed in this paper has a relatively better performance on organ boundary segmentation.

**Table 3.** Quantitative results of ours comparing with those of baseline on validation set(best).

| Organs | DSC(ours, %) | NSD(ours, %) | DSC(baseline, %) | NSD(baseline, %) |
|---|---|---|---|---|
| Liver | 93.73 | 91.30 | 95.04 | 93.62 |
| RK | 87.49 | 86.44 | 77.96 | 77.24 |
| Spleen | 92.78 | 91.40 | 79.95 | 79.66 |
| Pancreas | 76.94 | 87.52 | 66.20 | 76.71 |
| Aorta | 91.55 | 95.96 | 91.09 | 95.40 |
| IVC | 81.27 | 82.04 | 75.86 | 75.44 |
| RAG | 68.84 | 85.49 | 60.33 | 75.81 |
| LAG | 68.51 | 84.03 | 58.02 | 69.60 |
| Gallbladder | 69.19 | 71.00 | 42.50 | 40.56 |
| Esophagus | 75.16 | 84.87 | 65.46 | 76.80 |
| Stomach | 78.49 | 82.98 | 70.36 | 74.56 |
| Duodenum | 64.95 | 79.93 | 52.62 | 72.39 |
| LK | 81.94 | 83.68 | 81.67 | 81.98 |
| mean | 79.30 | 85.17 | 70.54 | 76.11 |

Compared with the segmentation results using only label data under the same experimental conditions, as shown in Table 3, the semi-supervised method proposed in this paper is improved on DSC, which directly demonstrate that effective utilization of unlabeled data can improve segmentation performance. The poor results using only labeled data also demonstrate the importance of the distribution of training data to model performance. Especially in the data of FLARE2022, there is the problem of different organ lesions between the training set and validation set, making their distributions vary and leading to extremely poor segmentation of some organs.



(a) Image          (b) Ground Truth          (c) Segmentation

**Fig. 2.** Qualitative evaluation of model performance on validation set. Row 1 and 2:Well-segmented examples. Row 3 and 4: challenging examples.

**Table 4.**  Quantitative results on testing set.

| Organs | DSC(%) | NSD(%) |
|---|---|---|
| Liver | 94.04 ± 8.33 | 93.37 ± 10.72 |
| RK | 87.22 ± 24.01 | 87.73 ± 24.54 |
| Spleen | 82.13 ± 28.72 | 83.14 ± 29.02 |
| Pancreas | 70.78 ± 22.99 | 81.01 ± 24.83 |
| Aorta | 91.56 ± 10.60 | 95.54 ± 11.32 |
| IVC | 79.67 ± 19.94 | 81.14 ± 20.74 |
| RAG | 73.67 ± 15. 07 | 90.76 ± 16.66 |
| LAG | 72.17 ± 17.95 | 87.16 ± 19.88 |
| Gallbladder | 69.93 ± 33.71 | 70.38 ± 34.54 |
| Esophagus | 68.93 ± 21.66 | 78.83 ± 23.77 |
| Stomach | 74.31 ± 23.30 | 76.89 ± 23.19 |
| Duodenum | 58.15 ± 25.13 | 74.25 ± 25.16 |
| LK | 88.60 ± 19.38 | 90.41 ± 19.20 |
| mean | 77.78 ± 11.58 | 83.89 ± 12.58 |

**Table 5.**  Segmentation efficiency results on testing set.

| Index | Time(s) | AUC CPU | AUC GPU |
|---|---|---|---|
|  | 18.70 ± 4.33 | 398.80 ± 77.67 | 25774.29 ± 11703.85 |

### 4.2   Qualitative results on validation set

Figure 2 presents the qualitative results on the validation set, where row 1 and 2 show some cases that are relatively easy to segment while row 3 and 4 show cases that are challenging to segment. It can be find that the cases of row 1 and 2 have clear boundaries and good contrast for the organs, and there exist no severe artifacts or lesions in the organs. Compared with the well-segmented cases, the challenge cases usually have noise(row 3) or organs lesion(row 4), which create difficulties in correctly segmenting the organ.

### 4.3   Results on final testing set

Table 4 shows the segmentation quantitative results on testing set. It can be seen that the results on testing set are very similar to those on the validation set, showing the robustness of our method. The running time and resource consumption are represented in Table 5. Compared with the results of each team finally displayed on the official website of the FLARE2022 competition[1], the method in this paper shows a relatively high efficiency.

## 5   Conclusion

Due to the rush of time in this work, in fact, only half a month was spent on research, the segmentation performance did not achieve our goal. In this work, it can be seen that the segmentation results of small organs is not very good. The deformation of the diseased organ is also not considered in the segmentation process. In the future, further research can be carried out in terms of boundary constraints and attention mechanisms.

**Acknowledgements** We declare that the segmentation method that implemented for participation in the FLARE 2022 challenge has not used any pretrained models nor additional datasets other than those provided by the organizers. The proposed solution is fully automatic without any manual intervention.

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
