# OpenReview forum: "Efficient Semi-Supervised Multi-Organ Segmentation Using Uncertainty Rectified Pyramid Consistency"
_MICCAI.org/2022/Challenge/FLARE_

### Official Review · Reviewer_8cF4 · 2022-09-12
**This paper has a strong resemblance with another last year MICCAI paper**

**Rating:** 1
**Confidence:** 5

**Review:**

This paper has a strong resemblance with another last year MICCAI paper: https://arxiv.org/pdf/2012.07042.pdf
Reviewer think that it might be coming from the same authors but this paper hardly mention the work in the previous one although it has no improvement or modification in the method (except the dataset).

Their (old) main contribution is to employ a pyramid structure into the 3D U-Net backbone, which outputs multi-level segment maps and optimize the uncertainty loss between them.

The reviewer does not see this paper as a legitimate paper since it looks the same as another work. In fact, many sentences are just copied from that work.

Some suggested improvement:
- Section 2.5 of the paper is redundant
- In page 7, the authors write that the proposed method has a relatively better performane on organ boundary segmentation. Some more explanation should be provided on this.
- Rewrite the whole paper, or at least paraphase those copied sentences.

---

### Official Review · Reviewer_uAHk · 2022-09-15
**The paper needs more explanation**

**Rating:** 6
**Confidence:** 3

**Review:**

The authors designed a semi-supervised method using multi-scale prediction with uncertainty rectified pyramid consistency loss in this work.

The semi-supervised method proposed in this paper significantly improves results by utilizing a large amount of unlabeled data.
There are some problems, which must be solved before it is considered for publication. If the following problems are well-addressed:

* Your contributions are not clearly outlined in the manuscript.
* The caption of the figures needs to be supplemented with a more detailed description.
* In Sections 2.1, 2.2, and 2.3, you used a large paragraph to describe your method and included a large number of variable definitions in the description. You should use appropriate figures to explain this and make the article more readable.

---

### Official Review · Reviewer_Znga · 2022-09-16
**Good idea, but it lacks a detailed description of the experiment.**

**Rating:** 6
**Confidence:** 4

**Review:**

Good paper, but it lacks a detailed description of the experiment.

**Summary:**

The authors introduce PPNet and uncertainty rectified pyramid consistency loss in this work for semi-supervised training. Compared with the baseline (fully-supervised 3D-UNet), the proposed method improves the Dice by about 3.5%.

**Strengths:**

- Table 4 indicated a great improvement for the Gallbladder, Esophagus, and Duodenum based on Pyramid Consistency. It's shown that the structure of this multi-level feature map is effective for small organs.

**Suggest improvements:**

- Author and institute are not given.
- Need more details for baseline experiment settings. Is the baseline network a plain 3D-UNet, like Fig. 1, without deep supervision and pyramid consistency?
- No descriptions for the inference stage. Is deep supervision available in the inference stage, fusing each decoder layer's output or just taking the last layer as output?
- Need more discussion about Table 4. Poor effect on small organ segmentation is a common problem, but notice that this method greatly improves small organs, which is more worth discussing.

---

### Official Review · Reviewer_k2u2 · 2022-09-16
**A semi-supervised Segmentation method using uncertainty rectified pyramid consistency**

**Rating:** 5
**Confidence:** 4

**Review:**

The paper uses 3D U-Net as a backbone network and adds an uncertainty rectifying module to the decoder for multi-organ segmentation.In general, the description of the network construction is relatively simple, and the overall framework of the network is not innovative, but the overall framework of the article is complete and the results presented on the validation set are good, but there are still some areas for improvement：
1、The Abstract does not explicitly state the results on validation set
2、You can introduce other people's methods in the introduction
3、Post-processing can be explained in more detail
4、It is recommended to place table4 and figure2 below the 4.2 text part, not in the middle of the paragraph

---

### Official Review · Reviewer_KBP5 · 2022-09-18
**The authors proposed an efficient semi-supervised framework with uncertainty rectified pyramid consistency regularization.  They extend a backbone to produce pyramid predictions for unlabeled images and encourage them to be consistent.  The multi-scale strategy is introduced to achive reliable segmentation results.**

**Rating:** 7
**Confidence:** 4

**Review:**

The authors uses pyramid prediction network to leverage the results of multi-scale feature maps.
For labeled images, the author uses multi-scale segmentation losses including Dice loss and CE loss.
For unlabeled images, the author uses L_2 norm to calculate the distance from prediction results and the mean prediction results.
Moreover, the authors select reliable voxels (with low uncertainty) to rectify the pyramid consistency loss for better leveraging the unlabeled images.
The method is very clear to understand.

1, The authors should provide the running time on the validation dataset, becaus FLARE2022 challenge is a joint low resource and accuracy oriented segmentation challenge, the authors should better provide the prediction time both in abstract and results as well as DSC.
2. CPU, RAM usage and GPU memory usage should be added in this paper, these results were send by the FLARE2022 organizers in the validation phase.
3. in section 4.1, the author should briefly introduce the reulsts, such as Aveage DSC is 77.98%. The key results should better added into the abstract.
4. In conclution part the authors claim that "Due to the rush of time in this work, in fact, only half a month was spent on
research, the segmentation performance did not achieve our goal." This sencence should better be removed, because it has nothing to do with the research.

---

### Official Review · Reviewer_9JFA · 2022-09-18
**In this paper, an efficient semi-supervised segmentation framework with uncertainty rectified pyramid consistency regularization is presented. in detail.**

**Rating:** 6
**Confidence:** 4

**Review:**

In this paper, a semi-supervised segmentation framework with uncertainty rectified pyramid consistency regularization is proposed, which can make full use of unlabeled data information to improve the segmentation efficiency. In this paper, a pyramid prediction network is introduced in detail for multi-scale prediction, and an uncertainty rectifying module is used to solve the problem of model collapse and detail loss.

Suggested Improvements:
The author needs to explain the segmentation efficiency strategy and show the segmentation efficiency index(running time and resource consumption).

---

### Public Comment · ~Zhengshan_Huang1 · 2022-09-21
**The article is clearly structured and contains all the necessary content**

The article is clearly structured and contains all the necessary content

---

### Meta-Review · Program_Chairs · 2022-09-28

**Recommendation:** Major Revision
**Confidence:** 5

**Metareview:**

Reviewers raise many concerns and suggestions. Please address all comments in the revised manuscript.